# Cannabidiol increases gramicidin current in human embryonic kidney cells: An observational study

**Mohammad-Reza Ghovanloo**[1,2¤a¤b], **Samuel J. Goodchild**[1], **Peter C. Ruben**[2]*

**1** Department of Cellular and Molecular Biology, Xenon Pharmaceuticals, Burnaby, BC, Canada,
**2** Department of Biomedical Physiology and Kinesiology, Simon Fraser University, Burnaby, BC, Canada

¤a Current address: Department of Neurology, Center for Neuroscience & Regeneration Research, Yale University School of Medicine, New Haven, CT, United States of America
¤b Current address: Neuro-Rehabilitation Research Center, Veterans Affairs Connecticut Healthcare System, West Haven, CT, United States of America
* pruben@sfu.ca

**Data Availability Statement:** All data are freely available without restriction at http://summit.sfu.ca/item/22375. The data are not ethically or legally restricted, nor are the data restricted by any ethics committee or IRB.

## Abstract

Gramicidin is a monomeric protein that is thought to non-selectively conduct cationic currents and water. Linear gramicidin is considered an antibiotic. This function is considered to be mediated by the formation of pores within the lipid membrane, thereby killing bacterial cells. The main non-psychoactive active constituent of the cannabis plant, cannabidiol (CBD), has recently gained interest, and is proposed to possess various potential therapeutic properties, including being an antibiotic. We previously determined that CBD's activity on ion channels could be, in part, mediated by altering membrane biophysical properties, including elasticity. In this study, our goal was to determine the empirical effects of CBD on gramicidin currents in human embryonic kidney (HEK) cells, seeking to infer potential direct compound-protein interactions. Our results indicate that gramicidin, when applied to the extracellular HEK cell membrane, followed by CBD perfusion, increases the gramicidin current.

## Introduction

Linear gramicidins are a family of antibiotics whose function is determined by increasing the cationic permeability of the membrane [1, 2]. Increased permeability is achieved by the formation of bilayer-spanning channels via dimerization of two hemi-channels. Relative to the channels formed by other antibiotics, gramicidin (gA) is well-behaved, and forms channels that are cation selective. Gramicidin channels are also among the best-understood of these types of channels. Atomic-resolution structures have been provided, and a wealth of functional experiments have yielded important insights into gA function [3, 4]. Gramicidin channel monomers that reside in each membrane leaflet must dimerize with monomers in the other leaflet to form a continuous pore. This dimerization change is necessary and sufficient for cationic currents to be conducted through gA. The channel association/dissociation and the related energetic

**Funding:** Natural Science and Engineering
Research Council of Canada and the Rare Disease
Foundation to PCR and M-RG (CGS-D: 535333-
2019 & MSFSS: 546467-2019), a MITACS
Accelerate fellowship in partnership with Xenon
Pharma, Inc. to M-RG (IT10714).

**Competing interests:** The authors have declared
that no competing interests exist.

cost comes from membrane deformation. The pore diameter is ~4 Å, sufficient to allow the
pore to also conduct alkali metals, protons, and water [2, 5–7].

The rate of gramicidin channel dimerization is directly related to membrane stiffness or
elasticity [4, 8]. This property has been the foundation of functional assays designed to deter-
mine the effects of various compounds on membrane dynamics. For example, compounds
that reduce the membrane stiffness or thickness (e.g. detergents) enhance the probability of
gramicidin dimerization, which in turn increases the cationic gramicidin signal [8–11].

Amphiphiles are among the compounds characterized using the gramicidin-based assays
[10]. Amphiphilic compounds are a set of molecules possessing both lipophilic and hydro-
philic properties. These molecules often display non-selective modulatory effects on seemingly
unrelated targets, a by-product of amphiphiles modulating membrane elasticity [8–10]. Modu-
lation is achieved when amphiphiles localize at the solution–bilayer interface, which is made
possible by the compounds' polar group residing at the interface with the hydrophobic region,
which then inserts into the bilayer core. Partitioning into the lipid bilayer alters membrane
elasticity, and changes phase preference and membrane curvature [8–10].

One compound with amphiphilic properties is cannabidiol (CBD), the primary non-psy-
chotropic constituent of *Cannabis sativa* [12]. CBD is a clinically and experimentally substanti-
ated therapeutic compound with efficacy against a variety of conditions, including seizure
disorders (for which CBD is FDA-approved), pain, and muscle spasms [13–19]. Furthermore,
CBD has been suggested to have antibiotic properties [20, 21]. Unlike the psychotropic Δ9-tet-
rahydracannabinol (THC), CBD has little to no affinity for endocannabinoid receptors [22,
23]. However, many studies have shown that CBD interacts with a wide range of other targets,
including a diverse array of ion channels [13, 14, 16, 17, 19, 24–26]. We previously character-
ized the full inhibitory effects of CBD on voltage-gated sodium channels (Nav) and deciphered
the mechanism through which CBD inhibits Nav currents [17, 24, 25, 27]. We further found
that an important component of this mechanism involves CBD altering membrane elasticity,
which was measured using a gramicidin-based fluorescence assay (GFA) [11, 24].

GFA is based on the gramicidin permeability to $Tl^+$, a quencher of the water-soluble fluoro-
phore 8-aminonaphthalene-1,3,6-trisulfonate (ANTS), encapsulated in large unilamellar vesi-
cles (LUVs) doped with gramicidin. The rate of $Tl^+$ influx, measured as the rate of
fluorescence quench, indicates the time-averaged number of gramicidin channels in the LUV
membrane [11]. Molecules that alter the thickness and elasticity of the LUV membrane also
alter the lipid bilayer contribution to the free energy of dimerization and, thus, the free energy
of dimerization [28]. Our previous findings in LUVs suggested that CBD decreases gramicidin
signals in that assay [24].

We sought to further characterize CBD effects on gramicidin currents using an electrophys-
iological HEK cell-based assay. In the present study, we investigated the interactions between
gramicidin and CBD over short exposures, using voltage-clamped human embryonic kidney
(HEK-293) cells in the absence and presence of gramicidin. In this purely observational study,
we report that in contrast to the GFA assay, CBD increases the gramicidin current in HEK
cells.

## Methods

### Cell culture

Suspension Human Embryonic Kidney (HEK-293) cells were used for automated patch-clamp
experiments. All cells were incubated at 37 ˚C/5% $CO_2$. All cell culture reagents were pur-
chased from ThermoFisher Scientific, Waltham, MA, unless otherwise noted.

## Patch-clamp

Automated patch-clamp recording was performed on untransfected HEK (CLS Cat# 300192/ p777_HEK293, RRID:CVCL_0045; ATCC, Manassas, VA, USA) cells. Currents were measured in the whole-cell configuration using a Qube-384 *(Sophion A/S, Copenhagen, Denmark)* automated voltage-clamp system. Intracellular solution contained (in mM): 120 KF, 10 NaCl, 2 $MgCl_2$, 10 HEPES, adjusted to pH7.2 with CsOH. The extracellular recording solution for the high sodium experiment contained (in mM): 140 NaCl, 3 KCl, 1 $MgCl_2$, 1.5 $CaCl_2$, 10 HEPES, adjusted to pH7.4 with NaOH. For the low sodium experiment the external solution sodium concentration was lowered to 1 mM with N-methyl-D-glucamine (NMDG) as NaCl replacement. Liquid junction potentials calculated to be ~7 mV were not adjusted for. Currents were low-pass-filtered at 5 kHz and recorded at 25 kHz sampling frequency. Series resistance compensation was applied at 100%. The measurements were obtained at room temperature which corresponds to $27 \pm 2$ ˚C at the recording chamber. Appropriate filters for cell membrane resistance (typically >500 M$\Omega$) and series resistance (<10 M$\Omega$) were used. We made fresh gramicidin stock from powder (5 mg/100 μL) in DMSO (26.56 mM). Gramicidin was dissolved in 100% DMSO, and the final concentration of 26 μM. CBD was purchased from Cayman Chemicals (No. 90080) and gramicidin was obtained from Sigma-Aldrich (CAS 11029-61-1).

The Sophion Qube is an automated electrophysiology instrument that is blinded to cell selections and experimentation, and selection was performed in a randomized manner. All subsequent data filtering and analysis was performed in a non-biased manner, in which automated filters were applied to the entire dataset from a given Qube run.

Electrophysiological data analysis

The analysis of raw patch-clamp recordings was performed using the Sophion Analyzer. Graphing and additional analysis was done using the Prism GraphPad (Version 9) software.

## Statistics

A one-factor analysis of variance (ANOVA) or t-test were, when appropriate, were used to compare the mean responses. *Post-hoc* tests using the Tukey Kramer adjustment compared the mean responses between channel variants across conditions. A level of significance $\alpha = 0.05$ was used in all overall *post-hoc* tests, and effects with p-values less than 0.05 were considered to be statistically significant. All values are reported as means ± standard error of means (SEM) for *n* recordings/samples.

## Results

### CBD increases gramicidin signal in HEK cells in high extracellular sodium concentrations

Gramicidin channels preferentially conduct cationic (e.g., $Na^+$ and $K^+$) currents upon dimerization and pore formation [1, 2]. We measured cationic currents through dimerized gramicidin channels using whole-cell voltage-clamp of untransfected HEK cells in the absence and presence of 26 μM gramicidin applied to the extracellular side of the membrane. First, we measured gramicidin currents in standard high sodium [$Na^+$ = 140 mM] extracellular solution using a ramp protocol. We clamped the cell membranes at -80 mV, close to the $K^+$ equilibrium potential ($E_K^+$). Then, we hyperpolarized the cells to -120 mV and ramped the voltage to +50 mV, which is close to $E_{Na}^+$. We added the gramicidin and compound combinations after about 22 minutes, when the currents were stable, then gramicidin/compound combos were added for a 5-minute interval, and the measurements were taken at the end of the interval (**S1 Fig**).We show average gramicidin current density from the ratio of current amplitude to the

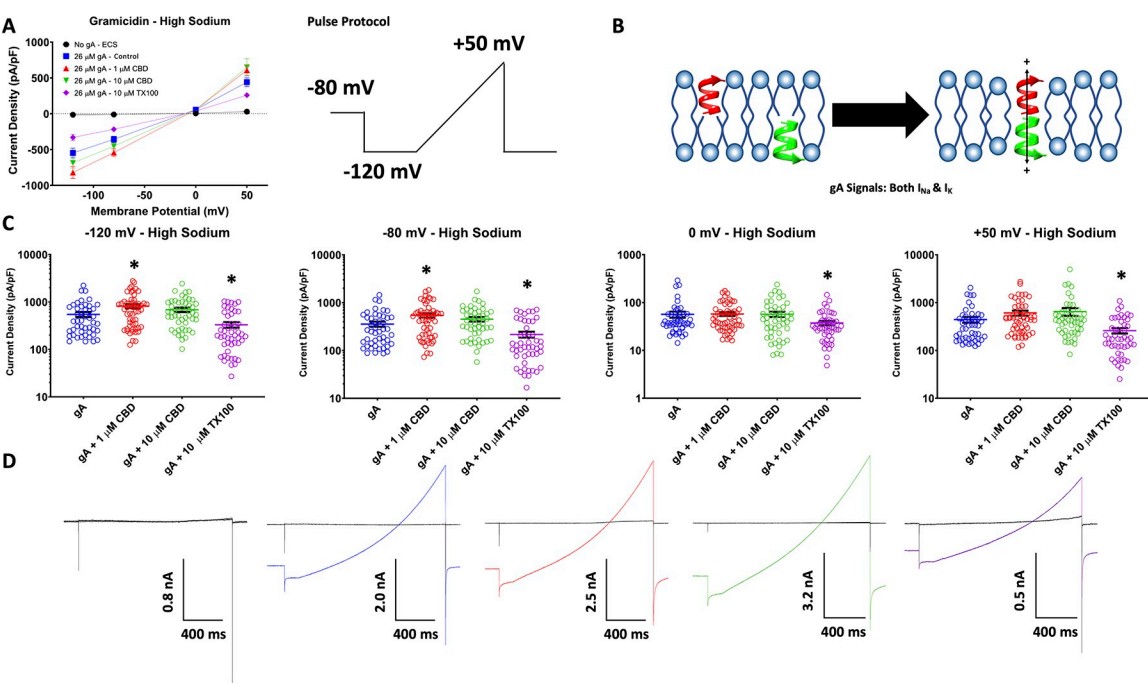

**Fig 1. High sodium voltage-clamp, gramicidin (gA).** (A) Shows the averaged cationic current densities of gramicidin in the presence/absence of CBD at 1 and 10 μM, and TX100 at 10 μM, on the left (in pA/pF, ECS: -120 mV = -13.6 ± 6.3, -80 mV = -10.1 ± 5.0, 0 mV = 5.2 ± 0.5, +50 mV = 30.0 ± 11.0, n = 36; gA: -120 mV = -547.8 ± 66.2, -80 mV = -357 ± 44, 0 mV = 56.5 ± 8.1, +50 mV = 441.1 ± 60.5, n = 47; 1 μM CBD: -120 mV = -820.1 ± 83.2, -80 mV = -543 ± 55.8, 0 mV = 57.7 ± 5.2, +50 mV = 607.4 ± 72.2, n = 56; 10 μM CBD: -120 mV = -687.0 ± 70.3, -80 mV = -452.8 ± 47.4, 0 mV = 57.1 ± 7.1, +50 mV = 649.2 ± 118.8, n = 49; 10 μM TX100: -120 mV = -330.8 ± 43.8 ±, -80 mV = -216.5 ± 30.4, 0 mV = 37.3 ± 4.0, +50 mV = 259.4 ± 32.8, n = 48). The ramp voltage protocol is shown on the right. (B) Shows a cartoon diagram of how gramicidin monomers are thought to dimerize and form channels. (C) Shows quantification of the data shown in (A), stars indicate statistical significance. (D) Shows the associated current traces.

cell membrane capacitance (pA/pF) at -120, -80, 0, and +50 mV (**Fig 1A–1D**). Our results indicate that, at negative potentials, gramicidin conducts inward currents and, as the membrane potential becomes more positive, the current becomes outward with the reversal potential ($E_{rev}$) being close to 0 mV, as would be predicted for a non-specific monovalent cationic channel. We also measured the effects of 1 μM and 10 μM CBD [17], and 10 μM Triton X100 (TX100; as positive control [9]) on gramicidin-HEK cells (**Fig 1A–1C**). TX100 is a detergent, and has been shown to change membrane elasticity and hence to increase gramicidin current amplitude in [9]. Interestingly, our findings indicate that TX100 reduced the cationic gramicidin currents across all potentials (-120 mV: p = 0.0118, -80 mV: p = 0.0123, +50 mV: p = 0.0311) (**Fig 1C**). CBD had the opposite effect to that of TX100, and slightly increased gramicidin currents at both 1 μM (-120 mV: p = 0.0152, -80 mV: p = 0.0136, +50 mV: p = 0.0368) and 10 μM (p>0.05). Interestingly, although the tendency for CBD to alter gramicidin currents was the same at both concentrations, CBD's effects were more variable at 10 μM than at 1 μM; this variability resulted in lack of statistical significance at 10 μM (**Fig 1C**). We speculate the variability at 10 μM may be due to damage to the HEK cell membrane from both gramicidin and CBD over the timescales of voltage-clamp experiments.

## CBD increases gramicidin signal in HEK cells in low extracellular sodium concentrations

The presence of a gramicidin dependent current indicates ion flux across the cell membrane. Gramicidin pores are analogous to puncturing cation-selective holes through the cell

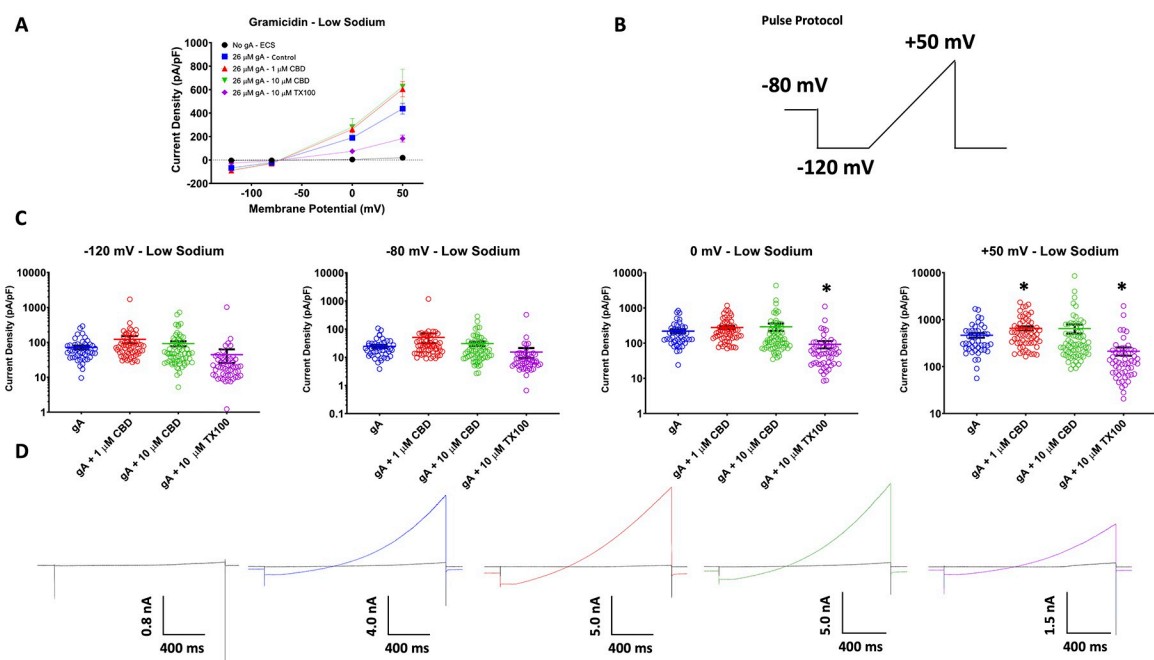

**Fig 2. Low sodium voltage-clamp.** Shows the averaged cationic current densities of gramicidin in the presence/absence of CBD at 1 and 10 μM, and TX100 at 10 μM (in pA/pF, ECS: -120 mV = -4.1 ± 0.9, -80 mV = -3.5 ± 0.7, 0 mV = 4.1 ± 0.4, +50 mV = 18.8 ± 1.7, n = 33; gA: -120 mV = -67.6 ± 7.7, -80 mV = -23.0 ± 2.9, 0 mV = 189.8 ± 20.6, +50 mV = 437.4 ± 45.6, n = 45; 1 μM CBD: -120 mV = -90.9 ± 9.0, -80 mV = -30.6 ± 2.9, 0 mV = 261.7 ± 28.7, +50 mV = 604.0 ± 64.8, n = 55; 10 μM CBD: -120 mV = -83.6 ± 14.1, -80 mV = -27.8 ± 5.2, 0 mV = 280.8 ± 74.4, +50 mV = 624.9 ± 149.6, n = 60; 10 μM TX100: -120 mV = -26.3 ± 3.5, -80 mV = -9.7 ± 1.3, 0 mV = 74.2 ± 11.3, +50 mV = 182.4 ± 29.0, n = 50). (B) Shows the ramp voltage protocol. (C) Shows quantification of the data shown in (A), stars indicate statistical significance. (D) Shows the associated current traces.

membrane. Gramicidin induced currents in the previous high [Na⁺] experiment resulted in a $E_{rev}$ close to 0 mV. This raises the possibility of a potential nonselective leak current component induced by gramicidin, but not carried by gramicidin, as a confounding variable. To ensure that we were recording gramicidin pore currents, we performed the same experiment with lower extracellular sodium [Na⁺ = 1 mM]. This experiment resulted in the same overall trends of altered gramicidin currents densities as the high [Na⁺] experiment, for both CBD and TX100 (**Fig 2A–2D**; **S1 Fig**). As expected, reducing [Na⁺] lowered the gramicidin $E_{rev}$ to ~-80 mV (close to $E_K^+$). These results confirm our results from the high Na⁺ experiment and further suggest that, when both Na⁺ and K⁺ are present at high concentrations, gramicidin permeability is not highly selective for K⁺ over Na⁺ bringing the gramicidin $E_{rev}$ to ~0 mV. Overall, these results show that CBD increases gramicidin currents during short exposures and suggests, therefore, that CBD could be altering membrane elasticity or gramicidin channel conductance directly.

## Discussion

In our previous study, using a GFA, it was determined that CBD has the opposite effect to TX100 and that it decreases the rate of dimerized gramicidin channel formation, and hence a smaller gramicidin current. These findings indicate that CBD is a modifier of the bilayer physical properties at the tested concentrations of 1–30 μM [24]. In this study, by electrophysiologically measuring K⁺ and Na⁺ currents flowing through the gramicidin channel, the opposite result was observed. CBD increased gramicidin currents and decreased TX100 currents

suggesting an alternate mechanism of gramicidin interaction with amphiphiles like CBD or TX100 than in the GFA assay.

Although the gramicidin structure does not indicate an obvious direct binding-site for CBD [3], there is a chance of a direct CBD-gramicidin interaction taking place. Indeed, in almost every report of CBD activity on a given target, a response has been determined, including various ion channels and receptor proteins [25, 29]. Therefore, the opposite result that was observed might suggest direct CBD and gramicidin interactions that in some way increase the probability of conducting pores in these conditions.

The molecular structure of CBD is composed of two oxygen atoms on both sides of a benzene ring, with the other two ends of the ring having a hydrocarbon tail on one end, and a hydrocarbon ring on the other. These features give the CBD molecule an overall shape that is loosely reminiscent of a phospholipid molecule. Phospholipids, in turn, are molecules that specialize in separating various cellular and sub-cellular environments, a function that is dependent on their amphiphilicity. In our previous paper, molecular dynamics (MD) simulations suggested that CBD molecules tend to localize below phospholipid headgroups, but above the tail-end region [24]. Thus, CBD molecules hovered around carbons ~3–7 of aliphatic chains, as per MD and verified by NMR [24]. It is conceivable that interactions between CBD positioned in the leaflet of the HEK membrane may interact with gramicidin hemi-channels to impact pore dimerization formation in a way that offsets any membrane stiffness affects that inhibit gramicidin currents, as we saw in previous GFA studies.

In the GFA assay, gramicidin monomers are incubated for 24 hours with liposomes at 13 ˚C to reach equilibration, and then the effect of compound is investigated by measuring fluorescence quenching rates [11]. In this study, we measured conventional macroscopic cationic currents in HEK cells using standard voltage-clamp, after the cells were extracellularly perfused with gramicidin monomers over the course of minutes at 27 ˚C. Therefore, the experimental setups between the two studies are fundamentally different, and likely investigate different phenomenon pertaining to gramicidin and CBD interactions. However, CBD's effect on gramicidin currents in this study appear to be relatively weak. A potential explanation for our results could also be that because we used gramicidin in the micromolar concentrations (instead of the nanomolar used in some studies), the gramicidin monomers adopted less canonical conformations. These conformations could be the reason for why CBD's effect, which is not very strong on gramicidin are such that the currents are increased [30]. Finally, there may also be direct interactions between TX100 and gramicidin monomers in our experiments.

Our goal in this study was to describe the effects of CBD on HEK cells externally treated with gramicidin. Our results suggest that there may be a direct interaction between CBD and gramicidin but the mechanism by which this potentiates gA currents in HEK cells remains unclear. Further studies will be required for instance, using MD simulations to examine potential for direct interactions.

## Supporting information

**S1 Fig. Time course of experiment.** (A) Shows the time course of experiment at high sodium and (B) at low sodium concentrations.
(DOCX)

## Author Contributions

**Conceptualization:** Mohammad-Reza Ghovanloo, Samuel J. Goodchild, Peter C. Ruben.

**Data curation:** Mohammad-Reza Ghovanloo, Samuel J. Goodchild.

**Formal analysis:** Mohammad-Reza Ghovanloo, Samuel J. Goodchild.

**Funding acquisition:** Peter C. Ruben.

**Investigation:** Mohammad-Reza Ghovanloo, Samuel J. Goodchild.

**Methodology:** Mohammad-Reza Ghovanloo, Samuel J. Goodchild.

**Project administration:** Peter C. Ruben.

**Resources:** Samuel J. Goodchild.

**Supervision:** Samuel J. Goodchild, Peter C. Ruben.

**Validation:** Samuel J. Goodchild.

**Writing – original draft:** Mohammad-Reza Ghovanloo.

**Writing – review & editing:** Samuel J. Goodchild, Peter C. Ruben.

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
