## [Decision Letter · Decision Letter 0]

21 Apr 2022

PONE-D-22-02483Cannabidiol increases gramicidin current in human embryonic kidney cells: An observational studyPLOS ONE

Dear Dr. Ruben,

Thank you for submitting your manuscript to PLOS ONE. After careful consideration, we feel that it has merit but does not fully meet PLOS ONE’s publication criteria as it currently stands. Therefore, we invite you to submit a revised version of the manuscript that addresses the points raised during the review process. If you decide to submit a suitably revised manuscript to PLOS One, please do so by Jun 05 2022 11:59PM. If you will need more time than this to complete your revisions, please reply to this message or contact the journal office at plosone@plos.org. Please include the following items when submitting your revised manuscript:A rebuttal letter that responds to each point raised by the academic editor and reviewer(s). You should upload this letter as a separate file labeled 'Response to Reviewers'.A marked-up copy of your manuscript that highlights changes made to the original version. You should upload this as a separate file labeled 'Revised Manuscript with Track Changes'.An unmarked version of your revised paper without tracked changes. You should upload this as a separate file labeled 'Manuscript'.

We look forward to receiving your revised manuscript.

Kind regards,

Mark S. Shapiro

Academic Editor

PLOS ONE

Journal Requirements:

"Natural Science and Engineering Research Council of Canada and the Rare Disease Foundation to PCR and M-RG (CGS-D: 535333-2019 & MSFSS: 546467-2019), a MITACS Accelerate fellowship in partnership with Xenon Pharma, Inc. to M-RG (IT10714)."

"Natural Science and Engineering Research Council of Canada and the Rare Disease Foundation to PCR and M-RG (CGS-D: 535333-2019 & MSFSS: 546467-2019), a MITACS Accelerate fellowship in partnership with Xenon Pharma, Inc. to M-RG (IT10714)."

3. We note that you have stated that you will provide repository information for your data at acceptance. Should your manuscript be accepted for publication, we will hold it until you provide the relevant accession numbers or DOIs necessary to access your data. If you wish to make changes to your Data Availability statement, please describe these changes in your cover letter and we will update your Data Availability statement to reflect the information you provide

Additional Editor Comments (if provided):

Dear Dr. Ruben:

Your manuscript has been reviewed by two specialists in your field, whose reviews are appended below. Both reviewers expressed concerns, but those of reviewer #2 are much more severe. If you feel that you can adequately address the concerns of both reviewers, especially #2, we would be happy to re-consider a substantially-revised manuscript for publication, which would be sent again to these two reviewers. Our apologies for the delay in having your submission reviewed. Thank you for submitting your work to PLOS One.

Sincerely yours,

Mark S. Shapiro, Ph.D.

Reviewing Editor

Reviewers' comments:

Reviewer's Responses to Questions

**Comments to the Author**

1. Is the manuscript technically sound, and do the data support the conclusions?

Reviewer #1: Partly

Reviewer #2: Partly

2. Has the statistical analysis been performed appropriately and rigorously? 

Reviewer #1: Yes

Reviewer #2: Yes

3. Have the authors made all data underlying the findings in their manuscript fully available?

Reviewer #1: Yes

Reviewer #2: Yes

4. Is the manuscript presented in an intelligible fashion and written in standard English?

Reviewer #1: Yes

Reviewer #2: Yes

5. Review Comments to the Author

Reviewer #1: This is a nice observational study of cannabidiol impacts on gramicidin in human embryonic kidney cells as a test of whether CBD has effects on bulk bilayer properties in mammalian cells. This is an interesting and important topic, as CBD is poorly soluble with very high partition coefficient and affects a wide array of membrane proteins. Whether CBD's promiscuity is generally due to bilayer effects or protein binding is unclear. The use of gramicidin in patch clamped cells to test for bilayer effects is smart. The study appears to be straightforward and observational as promised. This study will have value to readers interested int he topic of CBD-membrane interactions.

My major concern is that the data do not strongly support the conclusions. I suggest softening the conclusions. The CBD effect is weak, and I was not convinced that it is not a fluke of serial errors. The statement " CBD ... slightly increased gramicidin currents at ... 10 μM (p>0.05)." is logically flawed because the test indicates that the increase was not significant. The statistic of p<0.05 for 1 μM is not compelling. Additional statistical analyses and discussion of why the statistical test is appropriate could be helpful. Any information about how opportunities for serial errors were minimized between compound and cell prep could be important. A positive control for gA activation is lacking. I think this manuscript would better err on the side of caution and describe the weakness of the CBD effect, and discuss the possibility the effect could not even be real. I suggest changing the conclusion to "CBD does not appear to inhibit gA currents at concentrations where is active against other membrane proteins, and the statistical analyses suggest there may even be an activating effect."

Reporting transparency:

What specific gramicidin & CBD were used? Report source.

Report on care taken handling of gramicidin and CBD (tubes tubing etc). Both stick to surfaces and easily depleted from solution.

Report p-values rather than just p<0.05.

Report if blinding and randomization were implemented and how.

Report the sequence and timing of compound additions to HEK cells.

Please provide a metric for assessing what the maximum tolerated voltage error was. Series resistance compensation was applied at 100% yet the accuracy of the compensation is dependent on the accuracy of the series resistance estimate itself, and these estimates can be noisy. As current densities of up to 10 nA/pF are reported the potential for error seems real.

Please report whether series resistance was reassessed after every compound addition.

Reviewer #2: This manuscript by Ghovanloo Goodchild and Ruben describes the effects of CBD on gramicidin channels incorporated in HEK293 cells, using patch clamp.

Major comments

The description of the methods appears insufficient. In particular, application of gramicidin on only one side of the membrane would be difficult to compare to gramicidin added in the tiny amount used when added to both side of lipid bilayers. Did the authors thought of using a low concentration of gramicidin in the pipette to let the two sides bring a small amount of gramicidin as in the bilayer experiments of Andersen’s lab? How much was the stock in DMSO? How much DMSO was added to get to 26 μM?

It would be useful to see the full temporal course of the experiment. This is because the actions of gramicidin on biological membranes tend to take long to equilibrate when gramicidin is added on only one side (although this would depend on the concentration used and the biological membrane being considered). This will demonstrate that the drugs were always added only after the current is completely stable can the drugs be added. If this is not possible, at least give the time course of increase in conductance after application of gramicidin and how long after gramicidin application were the drugs added, and how long was the waiting time before measurements in the presence of drug.

I believe the large concentrations of gramicidin required to form pores in the whole cell are the reason why opposite than expected results are observed with triton. It would be good to discuss the differences between what is observed in HEK cells and what has been described in lipid bilayers and the possibility that at these massive concentrations part of the gramicidin channels are not as well behaved as in the bilayers. I think that it is good that the authors have the results in both systems. Perhaps what happens is that large part of the channels are not in the conformation normally seen at the low concentrations used in artificial lipid bilayers.

The gramicidin monomer/dimer association dissociation is not a conformational change. The channel associates and dissociates and the energetic cost for this only comes from the membrane deformation, please correct that statement.

About the discussion

I don’t think this results suggest that there is a different mechanism, I think this suggests that the channels formed by gramicidin are not the well behaved channels seen in artificial bilayers. This will require reviewing literature about different structures of gramicidin channels can adopt (I won’t dare to say this is a double stranded gramicidin, see Andersen et al NSB, 1999), but clearly gramicidin channel in artificial membranes form at concentrations of 10 nM not the 26 μM used in this work or to increase permeability in biological membranes. The possibility of the channel adopting strange conformations should be considered here, even if its cation selectivity remains.

As presented more than indicating that CBD acts through a different mechanism than bilayer modification, the data indicates that triton may be interacting directly with the gramicidin channel. I say this, because the observation with CBD is an increase in current, as expected from the GFA. The effect of triton is not.

Minor comments.

3rd line of the discussion, please correct the statement on bilateral modification to

These findings indicate that CBD is a modifier of the bilayer physical properties at the tested concentrations of 1-30 μM ...

The room temperature appears to be very high (27C), I am just confirming this is not a typo occurring twice.

6. PLOS authors have the option to publish the peer review history of their article (what does this mean?). If published, this will include your full peer review and any attached files.

Reviewer #1: No

Reviewer #2: No

---

## [Author Response · Author response to Decision Letter 0]

29 Jun 2022

Reviewer #1: This is a nice observational study of cannabidiol impacts on gramicidin in human embryonic kidney cells as a test of whether CBD has effects on bulk bilayer properties in mammalian cells. This is an interesting and important topic, as CBD is poorly soluble with very high partition coefficient and affects a wide array of membrane proteins. Whether CBD's promiscuity is generally due to bilayer effects or protein binding is unclear. The use of gramicidin in patch clamped cells to test for bilayer effects is smart. The study appears to be straightforward and observational as promised. This study will have value to readers interested int he topic of CBD-membrane interactions.

We thank the reviewer for a positive and constructive assessment of our manuscript. 

My major concern is that the data do not strongly support the conclusions. I suggest softening the conclusions. The CBD effect is weak, and I was not convinced that it is not a fluke of serial errors. The statement " CBD ... slightly increased gramicidin currents at ... 10 μM (p>0.05)." is logically flawed because the test indicates that the increase was not significant. The statistic of p<0.05 for 1 μM is not compelling. Additional statistical analyses and discussion of why the statistical test is appropriate could be helpful. Any information about how opportunities for serial errors were minimized between compound and cell prep could be important. A positive control for gA activation is lacking. I think this manuscript would better err on the side of caution and describe the weakness of the CBD effect, and discuss the possibility the effect could not even be real. I suggest changing the conclusion to "CBD does not appear to inhibit gA currents at concentrations where is active against other membrane proteins, and the statistical analyses suggest there may even be an activating effect.

We thank the reviewer for raising these points and suggestions. The statistical tests that were used in this study are outlined in the Methods section, in which each drug condition was compared with the gA (no drug) condition. The automated patch clamp system that we used takes measurements, head-to-head over the time of a given experiment, and therefore, the effect of a given drug and/or concentration on a given cell was taken at the exact same time. We agree with the reviewer that the effect of CBD is not a strong one. We have added/reflected this onto the revised manuscript. This new addition is under the Discussion section.

Reporting transparency:

What specific gramicidin & CBD were used? Report source. 

This information has been added to the Methods section. CBD powder was purchased from Cayman Chemicals. Gramicidin was obtained Sigma.

Report on care taken handling of gramicidin and CBD (tubes tubing etc). Both stick to surfaces and easily depleted from solution. 

In this study, we used nonbinding plates and tubes for preparing and the experiments with CBD and gramicidin. This information has been added to the Methods section. 

Report p-values rather than just p<0.05. 

Done. 

Report if blinding and randomization were implemented and how. 

This info has been added to methods: The Sophion Qube is an automated electrophysiology instrument that is blinded to cell selections and experimentation, and selection was performed in a randomized manner. All subsequent data filtering and analysis was performed in a non-biased manner, in which automated filters were applied to the entire dataset from a given Qube run.

Report the sequence and timing of compound additions to HEK cells. 

Gramicidin was first perfused followed by 5 mins of waiting followed by CBD/TX100 addition. 

Please provide a metric for assessing what the maximum tolerated voltage error was. Series resistance compensation was applied at 100% yet the accuracy of the compensation is dependent on the accuracy of the series resistance estimate itself, and these estimates can be noisy. As current densities of up to 10 nA/pF are reported the potential for error seems real. 

The Qube uses a proprietary algorithm to perform 100% compensation, after accurately measuring series resistance, as outlined below. Information can be found here: https://sophion.com/products/rs-compensation-qube-384/

Please report whether series resistance was reassessed after every compound addition. The Sophion Qube actively measures series resistance at every experimental block. The compensation is applied at each measurement. 

Reviewer #2: This manuscript by Ghovanloo Goodchild and Ruben describes the effects of CBD on gramicidin channels incorporated in HEK293 cells, using patch clamp.

Major comments

The description of the methods appears insufficient. In particular, application of gramicidin on only one side of the membrane would be difficult to compare to gramicidin added in the tiny amount used when added to both side of lipid bilayers. Did the authors thought of using a low concentration of gramicidin in the pipette to let the two sides bring a small amount of gramicidin as in the bilayer experiments of Andersen’s lab? How much was the stock in DMSO? How much DMSO was added to get to 26 μM?

We thank the reviewer for a positive and constructive assessment of our manuscript. 

“We made fresh gramicidin stock from powder (5 mg/100 µL) in DMSO (26.56 mM). Gramicidin was dissolved in 100% DMSO, and the final concentration of 26 µM. CBD was purchased from Cayman Chemicals (No. 90080) and gramicidin was obtained from Sigma-Aldrich (CAS 11029-61-1).”

It would be useful to see the full temporal course of the experiment. This is because the actions of gramicidin on biological membranes tend to take long to equilibrate when gramicidin is added on only one side (although this would depend on the concentration used and the biological membrane being considered). This will demonstrate that the drugs were always added only after the current is completely stable can the drugs be added. If this is not possible, at least give the time course of increase in conductance after application of gramicidin and how long after gramicidin application were the drugs added, and how long was the waiting time before measurements in the presence of drug. 

We added a supplementary figure and new section on this, as follows:

“We added the gramicidin and compound combinations after about 22 minutes, when the currents were stable, then gramicidin/compound combos were added for a 5-minute interval, and the measurements were taken at the end of the interval (Fig S1).”

I believe the large concentrations of gramicidin required to form pores in the whole cell are the reason why opposite than expected results are observed with triton. It would be good to discuss the differences between what is observed in HEK cells and what has been described in lipid bilayers and the possibility that at these massive concentrations part of the gramicidin channels are not as well behaved as in the bilayers. I think that it is good that the authors have the results in both systems. Perhaps what happens is that large part of the channels are not in the conformation normally seen at the low concentrations used in artificial lipid bilayers. 

Done. We thank the reviewer for this point. We have added this to the discussion section. 

The gramicidin monomer/dimer association dissociation is not a conformational change. The channel associates and dissociates and the energetic cost for this only comes from the membrane deformation, please correct that statement. 

Done.

About the discussion

I don’t think this results suggest that there is a different mechanism, I think this suggests that the channels formed by gramicidin are not the well behaved channels seen in artificial bilayers. This will require reviewing literature about different structures of gramicidin channels can adopt (I won’t dare to say this is a double stranded gramicidin, see Andersen et al NSB, 1999), but clearly gramicidin channel in artificial membranes form at concentrations of 10 nM not the 26 μM used in this work or to increase permeability in biological membranes. The possibility of the channel adopting strange conformations should be considered here, even if its cation selectivity remains. 

Done. We have added a section on the possibility of other conformations of gramicidin. 

As presented more than indicating that CBD acts through a different mechanism than bilayer modification, the data indicates that triton may be interacting directly with the gramicidin channel. I say this, because the observation with CBD is an increase in current, as expected from the GFA. The effect of triton is not. 

Done. We added this important point to the manuscript. 

Minor comments.

3rd line of the discussion, please correct the statement on bilateral modification to

These findings indicate that CBD is a modifier of the bilayer physical properties at the tested concentrations of 1-30 μM ... 

Done.

The room temperature appears to be very high (27C), I am just confirming this is not a typo occurring twice. 

Yes, the temperature on our patch-clamp setup was 27 degrees.

---

## [Editor Report · Decision Letter 1]

8 Jul 2022

Cannabidiol increases gramicidin current in human embryonic kidney cells: An observational study

PONE-D-22-02483R1

Dear Dr. Ruben,

We’re pleased to inform you that your manuscript has been judged scientifically suitable for publication and will be formally accepted for publication once it meets all outstanding technical requirements.

Kind regards,

Mark S. Shapiro

Academic Editor

PLOS ONE

Additional Editor Comments (optional):

I believe the authors have satisfactorily addressed the comments and concerns of the reviewers.
---

## [Editor Report · Acceptance letter]

14 Jul 2022

PONE-D-22-02483R1 

Cannabidiol increases gramicidin current in human embryonic kidney cells: An observational study 

Dear Dr. Ruben:

I'm pleased to inform you that your manuscript has been deemed suitable for publication in PLOS ONE. Congratulations! Your manuscript is now with our production department. 

Kind regards, 

on behalf of

Dr. Mark S. Shapiro 

Academic Editor

PLOS ONE